# Using Ballistocardiogram and Impedance Plethysmogram for Minimal Contact Measurement of Blood Pressure Based on a Body Weight-Fat Scale

**DOI:** 10.3390/s23042318

**Published:** 2023-02-19

**Authors:** Shing-Hong Liu, Yan-Rong Wu, Wenxi Chen, Chun-Hung Su, Chiun-Li Chin

**Affiliations:** 1Department of Computer Science and Information Engineering, Chaoyang University of Technology, Taichung City 41349, Taiwan; 2Biomedical Information Engineering Laboratory, The University of Aizu, Aizu-Wakamatsu City 965-8580, Japan; 3Institute of Medicine, School of Medicine, Chung-Shan Medical University, Taichung City 40201, Taiwan; 4Department of Internal Medicine, Chung-Shan Medical University Hospital, Taichung City 40201, Taiwan; 5Department of Medical Informatics, Chung-Shan Medical University, Taichung City 40201, Taiwan

**Keywords:** ballistocardiogram, impedance plethysmogram, pulse transit time (PTT), weight-fat scale, blood pressure

## Abstract

Electronic health (eHealth) is a strategy to improve the physical and mental condition of a human, collecting daily physiological data and information from digital apparatuses. Body weight and blood pressure (BP) are the most popular and important physiological data. The goal of this study is to develop a minimal contact BP measurement method based on a commercial body weight-fat scale, capturing biometrics when users stand on it. The pulse transit time (PTT) is extracted from the ballistocardiogram (BCG) and impedance plethysmogram (IPG), measured by four strain gauges and four footpads of a commercial body weight-fat scale. Cuffless BP measurement using the electrocardiogram (ECG) and photoplethysmogram (PPG) serves as the reference method. The BP measured by a commercial BP monitor is considered the ground truth. Twenty subjects participated in this study. By the proposed model, the root-mean-square errors and correlation coefficients (*r*^2^s) of estimated systolic blood pressure and diastolic blood pressure are 7.3 ± 2.1 mmHg and 4.5 ± 1.8 mmHg, and 0.570 ± 0.205 and 0.284 ± 0.166, respectively. This accuracy level achieves the C grade of the corresponding IEEE standard. Thus, the proposed method has the potential benefit for eHealth monitoring in daily application.

## 1. Introduction

The World Health Organization (WHO) released guidance for digital health according to the review of benefits, harms, acceptability, feasibility, resource use, and equity considerations of digital health interventions [1]. Digital health covers electronic health (eHealth) and mobile health (mHealth), both of which have been considered as approaches to fighting the Coronavirus Disease 2019 (COVID-19) pandemic [2,3]. The field of mHealth includes telecare, telehealth, telemedicine, mobile technology, and the internet of things (IoT). The field of eHealth comprises not only communication techniques in the health field, but also includes healthcare services and health monitoring [4,5]. Thus, digital health focuses on the user, collecting data in real-time from social activities, and using sophisticated analysis to extract knowledge from these data sources as a means of improving public health and providing basic services [6].

Some studies have proposed that engaged patients could use sophisticated digital technologies for self-monitoring and self-care [7,8]. Large amounts of data and information are measured from such digital apparatuses [9], relating to the past, present, and future physical and mental health or condition of an individual [10]. Apple and Google have recently developed health applications, such as Apple Health and Google Fit, for tracking fitness and health status [11]. These applications support interaction with various consumer apparatuses (mobile healthcare) for data integration, helping doctors remotely examine their patients’ overall health information. Users get to know the real-time status of their bodies and can adjust their activities or diet for healthy lifestyles [9].

In recent years, wearable devices have been widely used for mobile healthcare. Exercise watches or bands measure the heart rate, step count, and calorie expenditure [12,13]. An electrocardiogram (ECG) patch records a long-term ECG signal [14]; an electromyogram (EMG) patch detects muscle fatigue in real-time [15]; a reflective photoplethysmograph (PPG) measures oxygen saturation when exercising [16]; and a light reflection rheograph (LRR) examines deep vein thrombosis in the calf [17,18]. These wearable devices use IoT technology to transmit data to mobile platforms to accelerate interactive communication between users and healthcare providers. Therefore, development of novel or innovative digital physiological measurement methods will be a main focus for the popularization of eHealth.

When the heart pumps the blood once, a blood pulse in the aorta transmits to the peripheral arterioles. This transmission time is called the pulse transit time (PTT) and has a reciprocal relation to the pulse wave velocity (PWV). Bramwell and Hill [19] proposed a model to explain the relation between blood pressure (BP) and PWV according to the Moens-Korteweg equation [20]. Since the blood pulse is caused by the left ventricular contraction, the R wave of the electrocardiogram (ECG) is typically used as the starting time of PTT. The foot point of a pulse wave of peripheral vessels is considered the ending time of PTT. Therefore, Sharwood-Smith proposed the use of PTT to estimate continuous systolic BP for monitoring its instantaneous drop under anesthesia [21]. In recent years, many studies have explored techniques for cuffless BP measurement [22], some of which have been implemented in wearable devices for eHealth [23,24]. Thus, in this study, the ECG and PPG are used to estimate BP as the reference method.

Mechanical vibration of the aorta caused by heart contraction is transmitted to the body with multiple degrees of freedom, as measurable by mass sensors, such as strain gauge or accelerometer, at upper and lower limb locations [25,26]. Such a signal is the ballistocardiogram (BCG). The J wave of a BCG occurs just after isovolumetric contraction [27]. The RJ interval has a high relation with the pre-ejection period of the heart [28]. Thus, the J wave of the BCG could be considered the starting time of PTT. Shin et al. used BCG measured by a MEMS accelerometer placed at the wrist to replace ECG for BP measurement [29]. Martin et al. used a force sensor placed beneath the foot to acquire the BCG signal to estimate BP [30]. Liu et al. used a weight scale and PPG probe placed on the toe for cuffless BP measurement when users stand on the weight scale, exploring the PTT difference between the measurement methods by ECG and PPG on the finger, and BCG and PPG on the toe [31]. They found that the delay time between ECG and BCG was 82.8 ± 22.73 ms, and 61.6 ± 17.47 ms between PPGs on the finger and toe. The root-mean-square errors (ERMS) of estimated systolic blood pressure (SBP) and diastolic blood pressure (DBP) were 6.7 ± 1.6 mmHg and 4.8 ± 1.5 mmHg, respectively.

Bioelectrical impedance is a measurable electrical reactance of the ionic conduction within a body segment, which has a close relation with the volume of this conductor [32]. The fat mass (a non-conductor) of the human body is equal to the difference between body weight and fat-free mass [33]. The fat-free mass can be measured by a whole-body impedance measurement with foot-to-foot [34,35] or hand-to-foot methods [36]. The impedance plethysmography (IPG) is the change of reactance when blood flows [37], which shares the same physical characteristics as that of the PPG. Liu et al. used IPG measured on the forearm and ECG to perform cuffless BP measurement [38]. Some studies used a commercial bathroom scale to measure the BCG and IPG by strain gauges and footpads for the heartbeat detection [39,40]. In the IPG measurement, an alternate current source, 10 kHz and 1 mA_RMS_, was used to inject through two footpad electrodes beneath the subject’s toes.

The body weight-fat scale and blood pressure monitor are the most popular apparatuses in eHealth [41,42,43,44]. Many studies show that there is a positive relationship between being overweight or obese and hypertension [45,46]. Thus, daily measurements of body weight and BP are an important issue for self-care. To encourage the habit of self-monitoring, the apparatus should be easy and comfortable to use. Thus, the goal of this study is to explore minimal contact BP measurement based on a commercial body weight-fat scale when users stand on it. The PTT values were detected from BCG and IPG signals extracted from four strain gauges (SGs) and four footpads. Four PPT models for estimating BP were used to explore the reliability and reproducibility of the proposed method. There are three contributions in this study. First, The BP measurement system was integrated into a commercial body weight-fat scale. The driving current source for IPG measurement was supported by the body weight-fat scale. Second, the circuit for the IPG measurements were supported in the method sector. Third, the accurate differences of cuffless BP measured by the proposed and reference methods were explored. The delay time between PPG and IPG on the finger and toe was analyzed. Thus, the proposed method does not require the installation of any sensor on the body, meaning that the measurement of BP is easy, unobtrusive, and very practical.

## 2. Materials and Methods

Figure 1 shows the structure diagram of the measurement system in this study. The commercial body weight-fat scale is the Omron HBF-371 (Osaka, Japan), which measures weight and body fat mass. We designed a four SGs circuit [31] and four footpads circuit to extract BCG and IPG signals from the body weight-fat scale, a portable acquisition device to sample these signals and transmit them to a notebook PC, and a graphic user interface (GUI) to display and record these signals in real-time. The portable acquisition device has eight measuring channels, a 3.0 Bluetooth module, and independent dual power supplies from a battery [47]. The two reference signals, ECG and PPG, were acquired, displayed, and recorded synchronously [31]. PTT_BCG+IPG_ and PTT_ECG+PPG_ values were extracted from the four signals. Then, four PTT models were used to estimate BP.

### 2.1. Impedance Plethysmography

In bioelectrical impedance analysis, the total longitudinal impedance of the lower limbs is expressed as
(1)$$R_{b}=\frac{R_{o}R_{n}}{R_{o}+R_{n}}$$
where *R_b_* is the total impedance that includes the static impedance, *R_o_*, and the alternating impedance for the blood flow pulse, *R_n_*, [32]. Sensing bioelectrical impedance voltage, *V_BIP_*, is defined as
(2)$$V_{BIP}=I_{A}R_{b}=I_{A}\frac{R_{o}R_{n}}{R_{o}+R_{n}}$$
where *I_A_* is the constant current. The changed *V_BIP_* (impedance plethysmograph, IPG) is proportional to *R_n_*. Figure 2 shows the schematic circuit of the IPG measurement. The 32 kHz sinusoidal signal, supported by the body weight-fat scale, is amplified by an inverting amplifier (U9A), which drives a constant current (U1A) of 2.5 mA and demodulates the IPG by the multiplier (U3). The alternating signal is applied to the body by two footpads (body 1 and body 4 terminals). An instrument amplifier (U2) is used to pick up the modulated signal through two receiving footpads (body 2 and body 3 terminals). Three 2nd-order lowpass filters (U4A, U5A, U6A) of cutoff frequency 10 Hz filter high frequency noises, while two 2nd-order highpass filters (U1B, U6B) of cutoff frequency 0.3 filter the wandering baseline, and the notch filter of cutoff frequency 60 Hz filters the noise of the power line. Finally, the baseline of the IPG signal is shifted to 1.5 V by an offset shift circuit (U7B).

### 2.2. Digital Signal Processing

All measured signals are filtered to remove the wandering baseline and high frequency noise with the 4th-order Butterworth bandpass filter of the 0.5 Hz to 10 Hz bandwidth. To reduce differences of phase lag among different signals, an 8th-order all-pass filter was designed to equalize the group delay within the passband. Figure 3 shows the ECG (blue), PPG (red), differential PPG (DPPG, magenta), BCG (black), and IPG (green), and differential IPG (DIPG, purple). The PTT measured by BCG and IPG signals was defined as the interval between the J wave of BCG and the foot point of IPG for PTT1, and PTT2 is defined as the interval between the J wave of BCG and the main peak of DIPG. The Pan-Tompkins method was utilized to detect the R wave of ECG [48]. The first zero-crossing points of DIPG and DPPG after the R wave were defined as the foot-point times of IPG and PPG. The J wave of BCG is the first peak after the R wave. Then the first peaks of DIPG and DPPG are detected following their first zero-crossing points. In Figure 3, the R wave of ECG, J wave of BCG, and main peaks of DIPG and DPPG are marked by black dots, as are the foot points of IPG and PPG. Figure 4 shows the raw (top) and filtered (button) IPG signals.

### 2.3. PTT Models for Blood Pressure Estimation

We utilized four models used by Liu et al. [31]:(3)BP=a×PTT1+b
(4)$$BP=a\times \ln\left(PTT1\right)+b$$
(5)BP=a×PTT1+b×HR+c
(6)$$BP=\frac{a}{PTT1}+\frac{b}{PTT2}+c\times HR+d$$
where *HR* is the heart rate. We used linear multi-dimension regression to parameterize these models [49]. In Equation (7) below, *x_ij_* is the input variable, *y_i_* is the target output variable, and *ŷ_i_* is the estimated variable. Additionally, *i* indexes the ith data, and *j* indexes the jth parameter, including PTT1, PTT2, and HR. The regression model is
(7)$$y_{i}=m_{0}+m_{1}x_{i1}+m_{2}x_{i2}+\ldots +m_{k}x_{ik}+e_{i}$$
where *e_i_* is the error, and *M* = [*m*_0_, *m*_1_, …, *m_k_*] is the coefficient vector of the model. The loss function (root-mean-square error), *E_RMS_*, is used to evaluate performance of this method:(8)$$E_{RMS}={\left(\frac{1}{n}\sum_{i=1}^{n}{\left(y_{i}-{\hat{y}}_{i}\right)}^{2}\right)}^{0.5}$$
where *n* is the number of data points. 

### 2.4. Statistic Analysis

The quantitative data is expressed as the mean ± standard deviation (SD). A two-tailed paired *t*-test is used to show the difference of two variables. A *p*-value of 0.05 or less is considered statistically significant. The cross-correlation coefficient, *r*^2^, is the quantity that gives the quality of least squares fitting to the original data,
(9)$$r^{2}=\frac{[n{\left(\sum xy)-(\sum x)\left(\sum y\right)\right]}^{2}}{\left[n\sum x^{2}-{\left(\sum x\right)}^{2}][n\sum y^{2}-{\left(\sum y\right)}^{2}\right]}$$

Furthermore, the precision and agreement between the ground-truth BP and the estimated BP by reference and proposed methods are compared using a Bland–Altman plot.

### 2.5. Experiment Protocol

Twenty subjects (12 males and 8 females) participated in this study. Their ages were 20.8 ± 1.0 years (from 22 to 19 years of age), their weights were 63.0 ± 16.4 kg (from 115 to 43 Kg), and their heights were 167.3 ± 7.9 cm (from 186 to 152 cm). This experiment was approved by the Research Ethics Committee of Chung Shan University Hospital (No. CS2-21194), in Taichung city, Taiwan.

Subjects were asked to rest for five minutes and fill out an informed consent form before the measurement session. The subjects’ information—including age, weight, height, medical treatment of illness, and health condition—was recorded. Subjects afflicted with arrhythmia or asthma were excluded from the experiment. Figure 5 shows a real photo from this study. The BP measured by a commercial blood pressure monitor (HM-7320, Omron, Osaka, Japan) is used as the ground-truth BP. The cuff is wrapped on the left upper arm. The probe of PPG is placed on the middle finger of right hand. Two electrodes of ECG are placed on the right and left arms to measure the ECG. The subject is asked to stand on a commercial body weight-fat scale (Omron HBF-371; Osaka, Japan), and follow the experimental procedure:Subjects stand on the body weight-fat scale to measure ECG, PPG, IPG, and BCG signals for five minutes, and they measure BP once as a baseline.Subjects run on a treadmill at a speed of about 6 km/h for at least three minutes, and 8 km/h for the next four minutes. If the SBP is not raised to 20 mmHg higher than the resting SBP, subjects are requested to run longer.Subjects stand on the commercial body weight-fat scale again, measuring ECG, PPG, IPG, and BCG signals for six minutes. The BP is measured once a minute when standing on the body weight-fat scale.Each measurement session requires about 18 min. Subjects are measured four times. The interval between any two measurement sessions is at least a week.

Table 1 shows the maximum and minimum ground-truth systolic and diastolic BPs of all subjects across the four measurements. Four signals were partitioned into segments every minute. The qualities of four signals were determined by manual selection in each segment. If any one of the four signals measured did not have good quality for at least 10 s, that segment would be ignored. The PTT1 values and PTT2 values were extracted from each heartbeat of four signals, which would be ranked in descending order. The average of inter-quartile range of PTT1 and PTT2 values represents the PTT1 and PTT2 within one minute. Table 1 shows the number (*N*) of segments for each subject. The total number of PTT data is 291. The maximum changed SBP and DBP are 48 mmHg and 29 mmHg, and the minimum changed SBP and DBP are 24 mmHg and 8 mmHg.

## 3. Results

The estimated BPs by Equations (3)–(6) based on the proposed and reference methods were separately compared to ground-truth BP. *E_RMS_* and *r*^2^ were used to describe performances of the two methods. Table 2 shows the results of estimated SBP and DBP with Equation (3). When using the reference method, the *E_RMS_*s are 7.3 ± 2.1 mmHg and 4.8 ± 1.8 mmHg, respectively. When using the proposed method, the *E_RMS_*s are 10.2 ± 2.2 mmHg and 5.2 ± 2.0 mmHg, respectively. The *r*^2^s of the reference method are 0.510 ± 0.272 and 0.204 ± 0.215, respectively. Then, the *r*^2^s of the proposed method are 0.163 ± 0.168 and 0.081 ± 0.106, respectively.

Table 3 shows the results of estimated SBP and DBP using Equation (4). When using the reference method, the *E_RMS_*s are 7.2 ± 1.8 mmHg and 4.7 ± 1.7 mmHg, respectively. However, when using the proposed method, the *E_RMS_*s are 9.9 ± 2.2 mmHg and 4.9 ± 1.9 mmHg, respectively. The *r*^2^s of the reference method are 0.538 ± 0.231 and 0.212 ± 0.192, respectively. Then, the *r*^2^s of the proposed method are 0.198 ± 0.201 and 0.126 ± 0.213, respectively.

Table 4 shows the results of estimated SBP and DBP with Equation (5). When using the reference method, the *E_RMS_*s are 6.2 ± 1.5 mmHg and 4.4 ± 1.5 mmHg, respectively. However, when using the proposed method, the *E_RMS_*s are 7.3 ± 2.1 mmHg and 4.5 ± 1.8 mmHg, respectively. The *r*^2^s of the reference method are 0.678 ± 0.168 and 0.321 ± 0.246, respectively. Then, the *r*^2^s of the proposed method are 0.570 ± 0.205 and 0.278 ± 0.290, respectively.

Table 5 shows the results of estimated SBP and DBP with Equation (6). When using the reference method, the *E_RMS_*s are 6.0 ± 1.7 mmHg and 4.1 ± 1.6 mmHg, respectively. When using the proposed method, the *E_RMS_*s are 7.2 ± 2.2 mmHg and 4.4 ± 1.9 mmHg, respectively. The *r*^2^s of the reference method are 0.716 ± 0.162 and 0.439 ± 0.292, respectively. Then, the *r*^2^s of the proposed method are 0.618 ± 0.203 and 0.370 ± 0.275, respectively.

According to the results presented in Table 5, Equation (6) has the best performance for the proposed method. Bland–Altman plots are used to analyze the agreement between two different measurement methods. Figure 6 shows a Bland–Altman plot of estimated SBPs by Equation (6) and ground-truth SBPs. Figure 6a represents the reference method whose limits of agreement are 10.6 mmHg and −10.4 mmHg, and Figure 6b represents the proposed method whose limits of agreement are 12.8 mmHg and −12.5 mmHg. Figure 7 shows a Bland–Altman plot of estimated DBPs by Equation (6) and ground-truth DBPs. Figure 7a represents the reference method whose limits of agreement are 7.3 mmHg and −7.2 mmHg, and Figure 7b represents the proposed method whose limits of agreement are 7.4 mmHg and −7.2 mmHg.

In Table 2, Table 3, Table 4 and Table 5 it can be seen that performance of the reference method was better than that of the proposed method. Therefore, in Table 6, we explored the difference of PTT1 and PTT2 between the reference and proposed methods. We used the paired *t*-test to compare the differences of PTT1 and PTT2 data between the reference and the proposed methods. PTT1_ECG_ (164.8 ± 22.0 ms) and PTT2_ECG_ (227.1 ± 28.5 ms) were significantly lower than PTT1_BCG_ (298.9 ± 80.6 ms) and PTT2_BCG_ (364.1 ± 89.8 ms).

Since the ECG and BCG separately belong to the electric and mechanical signals, there is a delay time (DT) between the two signals. Moreover, PPG and IPG are both plethysmograms. Given that the measured locations are different, there is also a DT between the two signals. In Figure 3, the ECG and PPG are the phase lead of BCG and IPG. Table 7 shows the delay time (DT_ECG-BCG_) between the ECG and BCG, and the delay time (DT_PPG-IPG_) between the PPG and IPG. The number of heartbeats used to calculate the PTT1 and PTT2 is 4042. Notably, DT_ECG-BCG_ (82.1 ± 20.3 ms) is significantly lower than DT_PPG-IPG_ (120.7 ± 40.3 ms). 

## 4. Discussion

The accuracy of cuffless BP measurement for individual subjects depends upon the reference methods (noninvasive or invasive method), the calibrated moments (immediately following the measurement, one day after, or one month after), and BP states (exercising, resting, or medication) [50,51]. Therefore, cuff-calibration BP measurement devices would yield small errors in estimating subsequent BP. However, because of the calibration, these results merely reflect inter-individual differences in the reference BP levels. For calibration-free BP measurement devices, higher accuracy of BP estimation depends on the small BP range of the participant cohort, and the overall results did not reveal the hemodynamic measurement of individual subjects. Thus, cuffless BP measurement devices are not recommended for clinical practice [52]. However, these innovative techniques for cuffless BP measurement could be suitable for tracking BP changes in users with a comfortable measurement. Tracking the long-term trend of BP is very important information in eHealth. This study’s innovation was using a commercial body weight-fat scale to perform the BP measurement. This technique has the advantage of convenience with minimal contact. 

The performance of BP measurement using the reference method did not mirror that of previous studies [51]. In Table 5, the *E_RMS_*s of SBP and DBP measured by the reference method are only 6.0 ± 1.7 mmHg and 4.1 ± 1.6 mmHg; their *r*^2^s are 0.716 ± 0.162 and 0.439 ± 0.292, respectively. There are three main issues here. Firstly, the BP of subjects was raised by the exercise, which changes depend on inter-individual differences. Thus, the maximum BP changes of SBP and DBP approached 48 mmHg and 29 mmHg, and the minimum BP changes were 24 mmHg and 8 mmHg. Secondly, the post-exercise BP decreases—that is, measured BP is dynamic. Therefore, the accuracy of a commercial BP monitor would also decrease [53]. Thirdly, the reference BP was repeatedly measured once a minute, and it was difficult to synchronize the PTT. Therefore, for the proposed method, the *E_RMS_*s of SBP and DBP are 7.2 ± 2.2 mmHg and 4.4 ± 1.9 mmHg, while the *r*^2^s are 0.618 ± 0.203 and 0.371 ± 0.275, as shown in Table 5. According to the standard developed by the Institute of Electrical and Electronics Engineers (IEEE) [54], the accuracy of the proposed method reaches a C grading. However, these results were very close to those of the reference method. 

Liu et al. developed an IPG ring to replace the PPG probe, extracting PTT from the ECG and IPG signals. The *r*^2^ between SBP and PTT is merely 0.528 [38]. Moreover, Liu et al. proposed a cuffless BP measurement with the BCG and PPG signals [31]. When a user stood on a weight scale, the BCG signal was measured from the strain gauge of the weight scale and the PPG signal was measured from the PPG probe placed on a toe. The *E_RMS_*s of SBP and DBP were 6.7 ± 1.60 mmHg and 4.8 ± 1.47 mmHg, and the *r*^2^s of SBP and DBP were 0.606 ± 0.142 and 0.284 ± 0.166. These results closely matched those of our proposed method. However, the advantage of our approach is that PTT is extracted from the strain gauges and foot pads of the body weight-fat scale. The disadvantage is that the signal-noise ratio (SNR) of IPG is lower than that of PPG. Thus, the number of heartbeats used in this study was 4042, somewhat fewer than 4364 [31], since a higher quality IPG signal was hard to acquire.

As seen in Table 6, PTT1_ECG_ and PTT2_ECG_ are significantly lower than PTT1_BCG_ and PTT2_BCG_. However, in Liu et al. [31], PTT1_ECG_ and PTT2_ECG_ were significantly higher than PTT1_BCG_ and PTT2_BCG_. The pulse waves detected by the foot pads of IPG, and the PPG probe placed on a toe should have a similar PTT in the two studies. Interestingly, however, the DT between PPG (finger) to IPG (leg) (120.7 ± 40.3 ms) was greater than the DT between PPG (finger) to PPG (toe) (61.6 ± 17.47 ms). In bioelectrical impedance measurements, the human body is divided into five inhomogeneous segments, two for the upper limbs, two for the lower limbs and one for the trunk [32]. The bioelectrical impedance includes the impedances of tissue and blood fluid [55]. Moreover, impedance of tissue is much higher than that of blood fluid. The leg-to-leg impedance of tissue bears a relation to the fat-free mass of one’s body, a property which has been exploited in the commercial fat scale. However, the blood fluid in the lower limbs not only has arterial blood flow, but also venous blood flow (rheography). Thus, the pulse wave detected by the foot pads of the IPG could not be considered a unique arterial pulse at a fixed distance. 

The main limitation of the proposed method is that subjects must stand in a stable position because the SNRs of IPG and BCG all are lower than PPG and ECG. Moreover, ECG and PPG are not sensitive to any shaking or swaying of the body. However, when users with sarcopenia or Parkinson’s disease stand on a commercial body weight-fat scale, stable BCG and IPG are difficult to measure. Some digital signal processing methods, such as the adaptive filter [56], empirical mode decomposition [15], or principal component analysis [57], could be used to remove motion artifacts from the BCG and IPG signals in order to overcome this problem. However, if users, such as those afflicted with diabetes, have poor circulation in their lower limbs, their IPG will not be measured reliably. Therefore, they are not likely to be suitable candidates for using this method to measure BP. However, one advantage of the proposed method is the ability to make BP measurement without installing any sensors on the body. 

## 5. Conclusions

This study used the PTT extracted from the BCG and IPG signals—measured using a commercial body weight-fat scale and a standing subject—to estimate BP according to four different models. In terms of the performance of the proposed method, the expected result was worse than that of the reference method. However, *E_RMS_*s of estimated SBP and DBP by the proposed method were 7.2 ± 2.2 mmHg and 4.4 ± 1.9 mmHg, and the *r*^2^s were 0.618 ± 0.203 and 0.371 ± 0.275. These results achieve a C grading according to the corresponding IEEE standard. Thus, this method could be implemented in a minimal contact system to easily measure BP in the daily life, which has the potential to benefit eHealth management in the future.

## Figures and Tables

**Figure 1 sensors-23-02318-f001:**
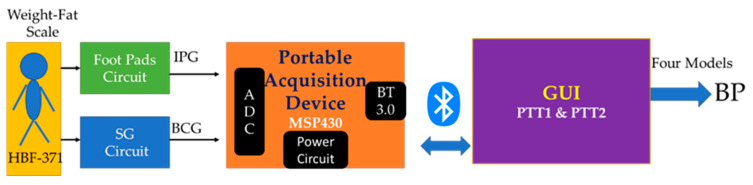
The structure of the minimal contact blood pressure measurement method.

**Figure 2 sensors-23-02318-f002:**
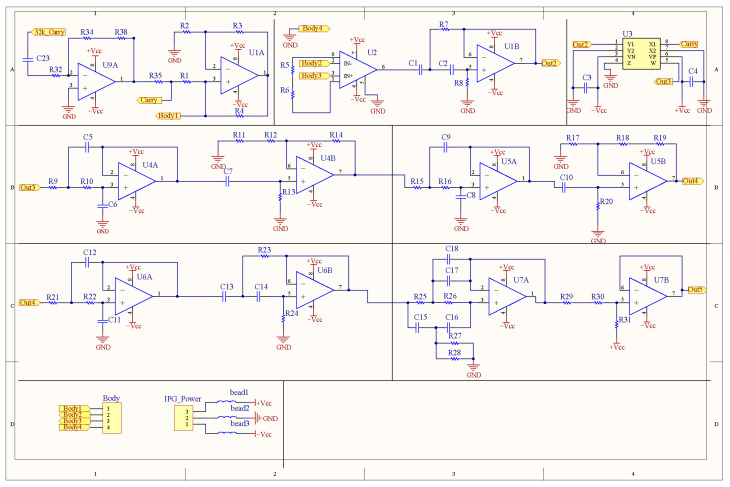
The circuit schematic of IPG measurement.

**Figure 3 sensors-23-02318-f003:**
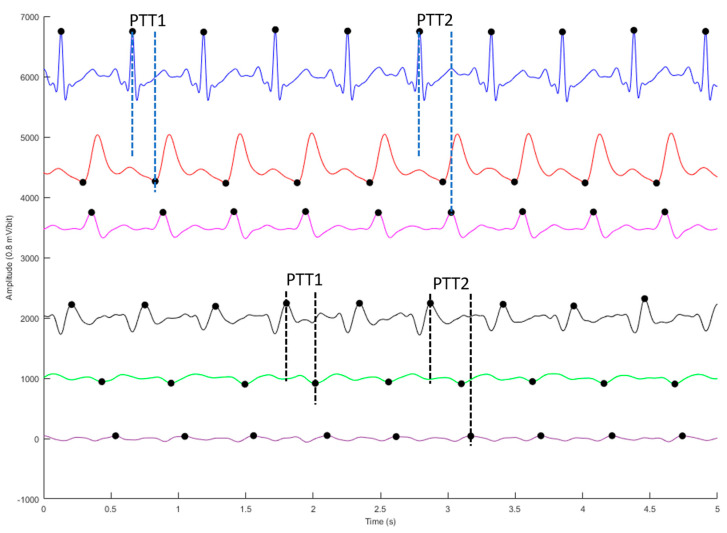
The four signals and their differential signals, ECG (blue), PPG (red), differential PPG (DPPG, magenta), BCG (black), IPG (green), and differential IPG (DIPG, purple). The R wave of ECG, J wave of BCG, main peak of DPPG and DIPG, and foot point of PPG and IPG are marked by black dots. These signals are offset to visually separate them.

**Figure 4 sensors-23-02318-f004:**
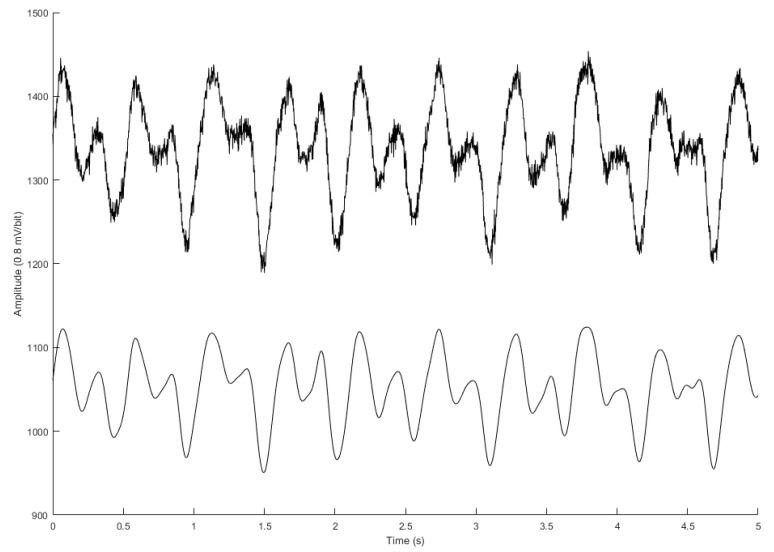
The raw (top) and filtered (bottom) IPG signals.

**Figure 5 sensors-23-02318-f005:**
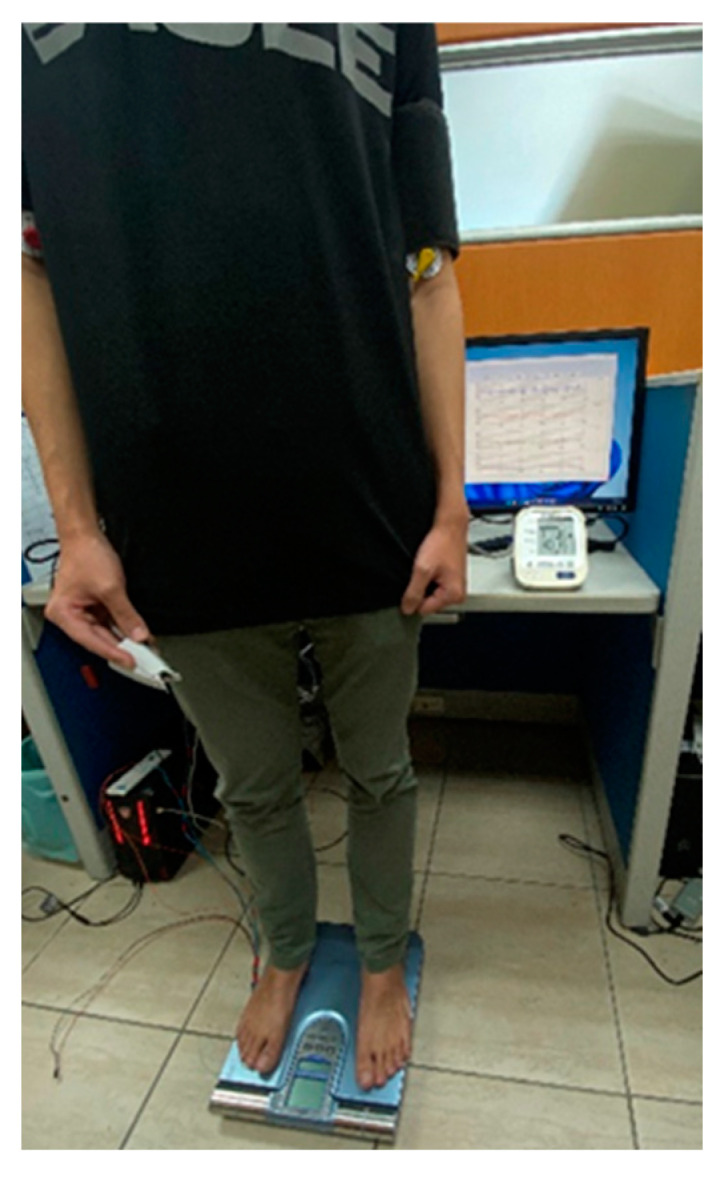
A photo from this experiment shows a subject standing on a commercial body weight-fat scale. The finger of right hand wears the PPG probe, a cuff is wrapped around the left arm, and two electrodes of the ECG are placed on the right and left arms.

**Figure 6 sensors-23-02318-f006:**
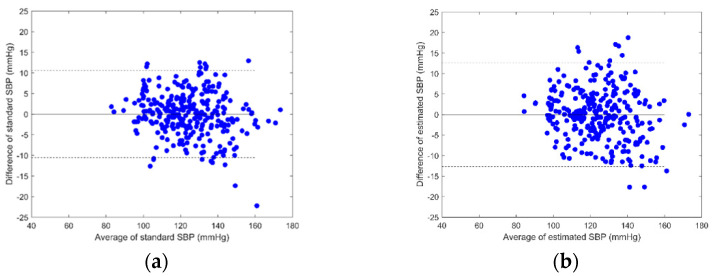
Bland–Altman plot of estimated SBPs by Equation (6) and ground-truth SBPs: (**a**) the reference method with ECG and PPG, (**b**) the proposed method with BCG and IPG.

**Figure 7 sensors-23-02318-f007:**
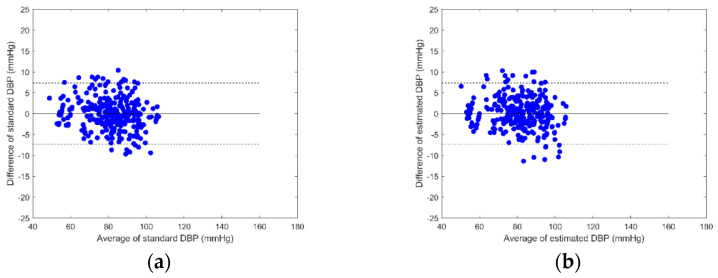
Bland–Altman plot of estimated DBPs by Equation (6) and ground-truth DBPs: (**a**) the reference method with ECG and PPG, (**b**) the proposed method with BCG and IPG.

**Table 1 sensors-23-02318-t001:** Maximum and minimum ground-truth systolic and diastolic BPs and number of useful segments for all subjects in this experiment.

Subject (*N*)	SBP Max.~Min. mmHg	DBP Max.~Min. mmHg	Subject (*N*)	SBP Max.~Min. mmHg	DBP Max.~Min. mmHg
01 (*N* = 15)	161~129	107~91	11 (*N* = 19)	172~124	107~85
02 (*N* = 17)	148~118	95~69	12 (*N* = 15)	146~105	88~59
03 (*N* = 19)	148~101	88~67	13 (*N* = 14)	140~112	88~76
04 (*N* = 17)	133~96	74~53	14 (*N* = 13)	128~98	88~76
05 (*N* = 14)	129~96	60~47	15 (*N* = 13)	137~105	90~80
06 (*N* = 13)	154~118	100~80	16 (*N* = 13)	129~82	94~68
07 (*N* = 15)	150~126	99~85	17 (*N* = 12)	147~116	78~70
08 (*N* = 14)	173~134	98~84	18 (*N* = 13)	163~120	106~87
09 (*N* = 12)	143~113	79~59	19 (*N* = 15)	133~96	85~74
10 (*N* = 15)	133~96	85~74	20 (*N* = 13)	126~84	87~69

ps: min. is the abbreviation of minimum, max. is the abbreviation of maximum.

**Table 2 sensors-23-02318-t002:** Results of estimated SBP and DBP with Equation (3) for the reference and proposed methods.

Subjects	Reference Method				Proposed Method			
	SBP		DBP		SBP		DBP	
	*E_RMS_* (mmHg)	*r* ^2^	*E_RMS_* (mmHg)	*r* ^2^	*E_RMS_* (mmHg)	*r* ^2^	*E_RMS_* (mmHg)	*r* ^2^
01	4.7	0.845	3.2	0.482	10.7	0.190	4.5	0.000
02	9.8	0.023	8.5	0.010	9.9	0.001	8.4	0.027
03	9.3	0.532	4.9	0.435	12.4	0.177	6.1	0.129
04	11.2	0.272	7.4	0.216	19.4	0.482	6.7	0.355
05	9.6	0.064	3.3	0.262	7.8	0.376	3.7	0.083
06	8.9	0.284	7.4	0.000	10.6	0.000	7.4	0.012
07	7.0	0.305	3.9	0.263	7.9	0.099	4.3	0.091
08	8.1	0.550	4.7	0.074	12.0	0.002	4.9	0.001
09	9.0	0.010	6.3	0.001	8.6	0.099	5.8	0.141
10	8.0	0.558	5.1	0.034	11.2	0.130	5.1	0.022
11	9.3	0.595	5.1	0.041	14.5	0.018	5.0	0.104
12	4.4	0.875	4.6	0.343	10.7	0.258	4.4	0.389
13	4.1	0.778	2.9	0.033	8.6	0.011	2.9	0.052
14	7.4	0.448	3.4	0.008	6.6	0.561	3.3	0.079
15	6.1	0.450	1.7	0.641	7.3	0.209	2.8	0.038
16	5.7	0.864	6.3	0.439	13.9	0.202	8.3	0.016
17	4.0	0.828	2.8	0.000	7.5	0.383	2.8	0.000
18	6.0	0.790	5.6	0.628	13.1	0.000	9.1	0.007
19	6.6	0.580	2.6	0.000	9.9	0.049	2.6	0.015
20	7.8	0.545	5.6	0.166	11.6	0.012	6.5	0.060
Mean ± SD	7.3 2.1	0.510 0.272	4.8 1.8	0.204 0.215	10.2 2.2	0.163 0.168	5.2 2.0	0.081 0.106

ps: SD, SBP, DBP are the abbreviations for standard deviation, systolic blood pressure, and diastolic blood pressure, respectively.

**Table 3 sensors-23-02318-t003:** Results of estimated SBP and DBP with Equation (4) for the reference and proposed methods.

Subjects	Reference Method				Proposed Method			
	SBP		DBP		SBP		DBP	
	*E_RMS_* (mmHg)	*r* ^2^	*E_RMS_* (mmHg)	*r* ^2^	*E_RMS_* (mmHg)	*r* ^2^	*E_RMS_* (mmHg)	*r* ^2^
01	6.0	0.751	3.5	0.385	10.7	0.200	4.5	0.001
02	9.7	0.044	8.4	0.031	9.6	0.057	8.0	0.127
03	9.1	0.552	4.9	0.435	13.0	0.089	6.5	0.014
04	9.8	0.441	6.3	0.433	5.7	0.807	3.0	0.871
05	9.1	0.146	3.4	0.236	7.2	0.463	3.5	0.161
06	8.4	0.363	7.4	0.009	10.5	0.009	7.4	0.004
07	6.9	0.310	3.9	0.246	7.8	0.132	4.4	0.030
08	7.8	0.578	4.7	0.085	12.1	0.000	4.9	0.001
09	8.0	0.229	5.4	0.263	8.5	0.117	5.9	0.114
10	6.9	0.669	5.0	0.060	11.1	0.138	5.2	0.015
11	9.4	0.586	5.2	0.016	14.0	0.084	5.0	0.073
12	5.6	0.800	4.6	0.318	11.5	0.137	3.8	0.531
13	4.3	0.755	3.0	0.011	8.4	0.053	2.9	0.038
14	7.3	0.452	3.4	0.010	7.1	0.489	3.2	0.117
15	6.0	0.468	1.9	0.557	8.1	0.029	2.8	0.033
16	4.5	0.915	5.9	0.511	13.0	0.301	8.4	0.000
17	4.1	0.820	2.8	0.001	7.3	0.425	2.8	0.000
18	5.8	0.800	6.7	0.459	11.1	0.275	8.9	0.050
19	6.9	0.532	2.6	0.000	9.7	0.073	2.6	0.027
20	7.8	0.546	5.6	0.178	11.2	0.079	5.1	0.318
Mean ± SD	7.2 1.8	0.538 0.231	4.7 1.7	0.212 0.192	9.9 2.2	0.198 0.201	4.9 1.9	0.126 0.213

ps: SD, SBP, DBP are the abbreviations for standard deviation, systolic blood pressure, and diastolic blood pressure, respectively.

**Table 4 sensors-23-02318-t004:** Results of estimated SBP and DBP with Equation (5) for the reference and proposed methods.

Subjects	Reference Method				Proposed Method			
	SBP		DBP		SBP		DBP	
	*E_RMS_* (mmHg)	*r* ^2^	*E_RMS_* (mmHg)	*r* ^2^	*E_RMS_* (mmHg)	*r* ^2^	*E_RMS_* (mmHg)	*r* ^2^
01	5.9	0.771	3.6	0.418	7.4	0.643	3.8	0.352
02	7.9	0.402	7.1	0.355	7.9	0.406	6.9	0.398
03	6.5	0.787	4.9	0.471	8.5	0.634	6.7	0.018
04	7.4	0.705	5.6	0.583	4.8	0.873	3.0	0.883
05	7.3	0.503	3.1	0.420	6.4	0.616	3.5	0.236
06	8.4	0.421	7.6	0.034	8.1	0.463	7.7	0.021
07	7.0	0.345	4.0	0.250	7.9	0.161	4.6	0.031
08	5.3	0.824	4.8	0.115	5.2	0.829	5.0	0.029
09	6.0	0.613	5.5	0.312	6.0	0.604	5.8	0.230
10	6.3	0.749	5.1	0.103	7.0	0.686	5.1	0.107
11	9.5	0.601	5.1	0.101	13.9	0.149	5.1	0.117
12	5.8	0.800	4.5	0.401	9.5	0.461	3.7	0.610
13	4.4	0.764	3.0	0.038	7.3	0.344	3.0	0.051
14	7.5	0.483	3.5	0.015	7.4	0.489	3.2	0.181
15	5.1	0.645	2.0	0.559	5.3	0.621	2.2	0.474
16	4.0	0.941	5.6	0.598	6.2	0.857	3.6	0.835
17	4.3	0.820	3.0	0.002	6.2	0.627	3.0	0.003
18	5.8	0.825	4.9	0.744	10.3	0.443	9.1	0.110
19	6.2	0.653	2.6	0.097	5.0	0.776	2.7	0.043
20	3.7	0.908	2.9	0.802	5.4	0.803	2.7	0.823
Mean ± SD	6.2 1.5	0.678 0.168	4.4 1.5	0.321 0.246	7.3 2.1	0.570 0.205	4.5 1.8	0.278 0.290

ps: SD, SBP, DBP are the abbreviations for standard deviation, systolic blood pressure, and diastolic blood pressure, respectively.

**Table 5 sensors-23-02318-t005:** Results of estimated systolic and diastolic blood pressures with Equation (6).

Subjects	Reference Method				Proposed Method			
	SBP		DBP		SBP		DBP	
	*E_RMS_* (mmHg)	*r* ^2^	*E_RMS_* (mmHg)	*r* ^2^	*E_RMS_* (mmHg)	*r* ^2^	*E_RMS_* (mmHg)	*r* ^2^
01	4.9	0.858	2.0	0.839	7.7	0.650	3.9	0.362
02	7.4	0.518	7.1	0.399	7.6	0.493	6.9	0.432
03	7.0	0.767	5.1	0.474	5.7	0.848	5.0	0.479
04	6.7	0.771	4.5	0.750	5.1	0.870	3.0	0.890
05	7.7	0.488	2.3	0.693	6.4	0.646	3.0	0.481
06	8.8	0.440	7.6	0.132	8.6	0.463	7.4	0.185
07	6.5	0.493	4.1	0.289	8.2	0.189	4.5	0.159
08	5.5	0.828	5.1	0.096	4.7	0.873	5.3	0.029
09	5.9	0.658	3.9	0.689	6.2	0.627	6.0	0.263
10	6.5	0.748	5.2	0.154	7.2	0.692	5.0	0.209
11	9.4	0.633	4.7	0.282	14.1	0.179	4.8	0.271
12	3.4	0.938	4.6	0.425	9.4	0.520	3.5	0.662
13	4.2	0.800	2.8	0.240	7.2	0.398	3.1	0.066
14	7.9	0.482	3.7	0.023	5.7	0.725	3.4	0.162
15	5.4	0.642	1.3	0.820	5.3	0.657	2.2	0.519
16	3.7	0.953	5.0	0.711	6.2	0.868	2.7	0.916
17	4.2	0.848	3.2	0.019	5.8	0.704	3.2	0.006
18	5.1	0.879	4.5	0.809	11	0.438	9.5	0.143
19	6.4	0.662	2.7	0.130	5.8	0.725	2.3	0.327
20	3.6	0.914	2.9	0.803	5.7	0.802	2.6	0.860
Mean ± SD	6.0 1.7	0.716 0.162	4.1 1.6	0.439 0.292	7.2 2.2	0.618 0.203	4.4 1.9	0.370 0.275

ps: SD, SBP, DBP are the abbreviations for standard deviation, systolic blood pressure, and diastolic blood pressure, respectively.

**Table 6 sensors-23-02318-t006:** The statistic of PTT1 and PTT2 measured by the reference and proposed methods.

	PTT1_ECG_ (ms)	PTT1_BCG_ (ms)	PTT2_ECG_ (ms)	PTT2_BCG_ (ms)
Mean	164.6	298.9	227.1	364.1
SD	22.0	80.6	28.5	89.8
*p*-value	0.0		0.0	

ps: SD is the abbreviation of standard deviation.

**Table 7 sensors-23-02318-t007:** The statistic of delay time (DT_ECG-BCG_) between the ECG and BCG, and delay time (DT_PPG-IPG_) between the PPG and IPG.

	DT_ECG-BCG_ (ms)	DT_PPG-IPG_ (ms)
Mean	82.1	120.7
SD	20.3	40.3
*p*-value	0.000	

ps: SD is the abbreviation of standard deviation.

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
