# Peer review of "Using Ballistocardiogram and Impedance Plethysmogram for Minimal Contact Measurement of Blood Pressure Based on a Body Weight-Fat Scale"

_sensors, 2023, doi:10.3390/s23042318_

Round 1

Reviewer 1 Report

The manuscript, Using Ballistocardiogram and Impedance Plethysmogram for Cuffless and Touchless Measurement of Blood Pressure Based on a Weight-Fat Scale, aims to demonstrate that a BCG and IPG-based measurement system is capable of tracking changes in systolic and diastolic pressures for home use. Such systems, due to their ease of use, could reduce the burden of using digital health technologies at home for personal use.  The manuscript clearly presents that although the results and capabilities of the proposed system are lower than existing methods, the proposed method could still be a viable at-home monitoring technique.

Main concerns with the manuscript:

The study's premise and investigation are that the proposed method is “Cuffless and Touchless.” The first is true; however, to measure IPGs, the system needs direct contact with the skin, so the system contacts/touches the user. Perhaps minimally invasive would be a better description of the system.

It is concerning how similar Figure 3 is to Figure 4 from “Cuffless and Touchless Measurement of Blood Pressure from Ballistocardiogram Based on a Body Weight Scale.”

In the background/introduction section, previous studies using BCG and IPG should be referenced, and then it should be clear what novelty this manuscript presents compared to them. For example, BCG and IPG were used by Park, Inan, and Giovangardi in “A Combined Heartbeat Detector based on Individual BCG and IPG Heartbeat Detectors.” Also see Inan, Park, Giovangradi, and Kovacs, “Noninvasive Measurement of Physiological Signals on a Modified Home Bathroom Scale.”

Lines 69 and 70 explain why the ECG R-wave can serve as the starting time for PTT. Lines 78 through 80 states, “The ballistocardiography (BCG) is the measurement of the recoil force of the body, which can be measured at upper and lower limb locations [25,26]. Thus, BCG also could be considered as the starting time of PTT.” The manuscript seems to argue that because the BCG could be measured at different site locations, it could be used as the starting point for PTT due to the signal having a time delay between the measurement sites. However, in this manuscript, both the BCG and IPG are measured at the same location – at the feet. More explanation should be provided on why the BCG, especially measured at the feet (whole-body BCG) can be used as the starting point for PTT.

Minor comments:

Lines 101 to 103 state, “In order to improve the habit of self-monitoring, the apparatus needs to be easy and comfortable to use. Thus, the goal of this study is to explore the cuffless and touchless BP measurement based on a commercial weight-fat scale when users standing on it.” More explanation should be provided on how the proposed system is “comfortable to use” especially compared to existing systems.

The text in Figure 2 is too small to read.

Figure 5 and 6 reference "(a)" and "(b"), but it is unclear what a and b are referencing.

Author Response

Reviewer 1 (round 1)

Dear Anonymous Reviewer,

The authors are grateful to your comments and suggestions for improving the quality and presentation of this paper. All comments are followed. The revised parts are highlighted in red. It is our sincere hope that this revision will enhance readability and strengthen of the manuscript to satisfy the requirements of this prestigious journal.

Comments and Suggestions for Authors

The manuscript, Using Ballistocardiogram and Impedance Plethysmogram for Cuffless and Touchless Measurement of Blood Pressure Based on a Weight-Fat Scale, aims to demonstrate that a BCG and IPG-based measurement system is capable of tracking changes in systolic and diastolic pressures for home use. Such systems, due to their ease of use, could reduce the burden of using digital health technologies at home for personal use.  The manuscript clearly presents that although the results and capabilities of the proposed system are lower than existing methods, the proposed method could still be a viable at-home monitoring technique.

Main concerns with the manuscript:

  1. The study's premise and investigation are that the proposed method is “Cuffless and Touchless.” The first is true; however, to measure IPGs, the system needs direct contact with the skin, so the system contacts/touches the user. Perhaps minimally invasive would be a better description of the system.

ANS: Thanks for Reviewer 1 and 3 comment and suggestion. Authors change the title of this manuscript as “Using Ballistocardiogram and Impedance Plethysmogram for Minimal Contact Measurement of Blood Pressure Based on a Body Weight-Fat Scale”.

  1. It is concerning how similar Figure 3 is to Figure 4 from “Cuffless and Touchless Measurement of Blood Pressure from Ballistocardiogram Based on a Body Weight Scale.”

ANS: The previous study, “Cuffless and Touchless Measurement of Blood Pressure from Ballistocardiogram Based on a Body Weight Scale”, used BCG and PPG to measure the blood pressure. But, in this study, the BCG and IPG were used to measure the blood pressure. Moreover, the ECG and PPG were the reference method. Thus, the differences between two figures are the IPG and PPG (from toe). But authors did a mistake for IPG and DIPG signals in Figure 3. The Fig. 3 has been modified. In order to let IPG signal more clear, Fig. 4 is added in the manuscript to show the raw and filtered IPG signals.

Line 150-176:

2.2 Digital Signal Processing

All measured signals are filtered to remove the wandering baseline and high frequency noise with the 4th-order Butterworth bandpass filter of the 0.5 Hz to 10 Hz bandwidth. To reduce differences of phase lag among different signals, an 8th-order all-pass filter was designed to equalize the group delay within the passband. Figure 3 shows the ECG (blue), PPG (red), differential PPG (DPPG, magenta), BCG (black), and IPG (green), and differential IPG (DIPG, purple). The PTT measured by BCG and IPG signals was defined as the interval between the J wave of BCG and the foot point of IPG for PTT1, and PTT2 is defined as the interval between the J wave of BCG and main peak of DIPG. The Pan and the Tompkins method was utilized to detect the R wave of ECG [48]. The first zero-crossing points of DIPG and DPPG after the R wave were defined as the foot-point times of IPG and PPG. The J wave of BCG is the first peak after the R wave. Then the first peaks of DIPG and DPPG are detected following their first zero-crossing points. In Figure 3, the R wave of ECG, J wave of BCG, and main peaks of DIPG and DPPG are marked by black dots, as are the foot points of IPG and PPG. Figure 4 shows the raw (top) and filtered (button) IPG signals.

Figure 3. The four signals and their differential signals, ECG (blue), PPG (red), differential PPG (DPPG, magenta), BCG (black), IPG (green), and differential IPG (DIPG, purple). The R wave of ECG, J wave of BCG, main peak of DPPG and DIPG, and foot point of PPG and IPG are marked by black dots. These signals are offset to visually separate them.

Figure 4. The raw (top) and filtered (bottom) IPG signals.

  1. In the background/introduction section, previous studies using BCG and IPG should be referenced, and then it should be clear what novelty this manuscript presents compared to them. For example, BCG and IPG were used by Park, Inan, and Giovangardi in “A Combined Heartbeat Detector based on Individual BCG and IPG Heartbeat Detectors.” Also see Inan, Park, Giovangradi, and Kovacs, “Noninvasive Measurement of Physiological Signals on a Modified Home Bathroom Scale.”

ANS: We have rewritten the texts of two paragraphs in “Introduction” sector according to reviewer comment.

Line 93-117:

Bioelectrical impedance is a measurable electrical reactance of the ionic conduction within a body segment, which has a close relation with the volume of this conductor [32]. The fat mass (a non-conductor) of human body is equal to the difference between body weight and fat-free mass [33]. The fat-free mass can be measured by whole-body impedance measurement with foot-to-foot [34, 35] or hand-to-foot methods [36]. The impedance plethysmography (IPG) is the change of reactance when blood flows [37], which physical characteristics are same as that of the PPG. Liu et al. used IPG measured on the forearm and ECG to perform cuffless BP measurement [38]. Some studies used a commercial bathroom scale to measure the BCG and IPG by strain gauges and footpads for the heartbeat detection [39,40]. In the IPG measurement, an alternate current source, 10 kHz and 1 mARMS, was used to inject through two footpad electrodes beneath the subject’s toes.

The body weight-fat scale and blood pressure monitor are the most popular apparatuses in eHealth [41–44]. Many studies show that there is a positive relationship between being overweight or obese and hypertension [45, 46]. Thus, daily measurements of body weight and BP are an important issue for self-care. To encourage the habit of self-monitoring, the apparatus should be easy and comfortable to use. Thus, the goal of this study is to explore minimal contact BP measurement based on a commercial body weight-fat scale when users stand on it. Users can easily measure their BPs by cuffless measurement. The PTT values were detected from BCG and IPG signals extracted from four strain gauges (SGs) and four footpads. The alternate current source for IPG measurement was supported by the commercial body weight-fat scale. To validate performance of the proposed method, the BP estimated by the reference method was compared with that of the proposed method. Four PPT models for estimating BP were used to explore the reliability and reproducibility of the proposed method.  The BP measurement system was integrated into a commercial body weight-fat scale. There is no need to install any sensor on the body, so measurement of BP is easy, unobtrusive, and very practical.

  1. D. Park; O. T. Inan; L. Giovangrandi. A combined heartbeat detector based on individual BCG and IPG heartbeat detectors. 2012 Annu. Int. Conf. IEEE. Eng. Med. Biol. Soc. 2012, 3428-31. doi: 10.1109/EMBC.2012.6346702.
  2. O. T. Inan; D. Park; L. Giovangrandi; G. T. A. Kovacs. Noninvasive measurement of physiological signals on a modified home bathroom scale. IEEE Tran. Biomed. Eng. 2012, 59, 2137-2143.

  1. Lines 69 and 70 explain why the ECG R-wave can serve as the starting time for PTT. Lines 78 through 80 states, “The ballistocardiography (BCG) is the measurement of the recoil force of the body, which can be measured at upper and lower limb locations [25,26]. Thus, BCG also could be considered as the starting time of PTT.” The manuscript seems to argue that because the BCG could be measured at different site locations, it could be used as the starting point for PTT due to the signal having a time delay between the measurement sites. However, in this manuscript, both the BCG and IPG are measured at the same location – at the feet. More explanation should be provided on why the BCG, especially measured at the feet (whole-body BCG) can be used as the starting point for PTT.

ANS: According to reviewer’s comment, we modified the description of the paragraph.

Line 77-91:

Mechanical vibration of the aorta caused by heart contraction is transmitted to the body with multiple degrees of freedom, as measurable by mass sensors, such as strain gauge or accelerometer, at upper and lower limb locations [25, 26]. Such a signal is the ballistocardiogram (BCG). The J wave of a BCG occurs just after isovolumetric contraction [27]. The RJ interval has high relation with the pre-ejection period of the heart [28]. Thus, the J wave of the BCG could be considered the starting time of PTT.  Shin et al. used BCG measured by a MEMS accelerometer placed at the wrist to replace ECG for BP measurement [29]. Martin et al. used a force sensor placed beneath the foot to acquire the BCG signal to estimate BP [30]. Liu et al. used a weight scale and PPG probe placed on the toe for cuffless BP measurement when users stand on the weight scale, and explored the PTT difference between the measurement methods by ECG and PPG on the finger, and BCG and PPG on the toe [31]. They found that the delay time between ECG and BCG was 82.8 ± 22.73 ms, and was 61.6 ± 17.47 ms between PPGs on the finger and toe. The root-mean-square error (ERMS) of estimated systolic blood pressure (SBP) and diastolic blood pressure (DBP) were 6.7 ± 1.6 mmHg and 4.8 ± 1.5 mmHg, respectively.

  1. I. Starr; H. A. Schroeder. Ballistocardiogram. II. normal standards, abnormalities commonly found in diseases of the heart and circulation, and their significance. J. Clinical Investigation. 1940, 19, 437-450.
  2. M. Etemadi; O. T. Inan; L. Giovangrandi; G. T. A. Kovacs. Rapid assessment of cardiac contractility on a home bathroom scale. IEEE Tran. Inf. Tec. Biomed. 2011, 15, 864-869.

Minor comments:

  1. Lines 101 to 103 state, “In order to improve the habit of self-monitoring, the apparatus needs to be easy and comfortable to use. Thus, the goal of this study is to explore the cuffless and touchless BP measurement based on a commercial weight-fat scale when users standing on it.” More explanation should be provided on how the proposed system is “comfortable to use” especially compared to existing systems.

ANS: According to the first comment, authors changed the title of this manuscript. Thus, this sentence also was modified to follow the contribution of this study, “minimal contact BP measurement”. 

Line 103-117

The body weight-fat scale and blood pressure monitor are the most popular apparatuses in eHealth [41–44]. Many studies show that there is a positive relationship between being overweight or obese and hypertension [45, 46]. Thus, daily measurements of body weight and BP are an important issue for self-care. To encourage the habit of self-monitoring, the apparatus should be easy and comfortable to use. Thus, the goal of this study is to explore minimal contact BP measurement based on a commercial body weight-fat scale when users stand on it. Users can easily measure their BPs by cuffless measurement. The PTT values were detected from BCG and IPG signals extracted from four strain gauges (SGs) and four footpads. The alternate current source for IPG measurement was supported by the commercial body weight-fat scale. To validate performance of the proposed method, the BP estimated by the reference method was compared with that of the proposed method. Four PPT models for estimating BP were used to explore the reliability and reproducibility of the proposed method.  The BP measurement system was integrated into a commercial body weight-fat scale. There is no need to install any sensor on the body, so measurement of BP is easy, unobtrusive, and very practical.  

  1. The text in Figure 2 is too small to read.

ANS: Authors enlarges the size of Fig. 2 to make it easier for readers to read the text in the figure.

  1. Figure 5 and 6 reference "(a)" and "(b"), but it is unclear what a and b are referencing.

ANS: Authors modified the captions of Fig. 6 and 7 to let their descriptions more clear.

Figure 6. Bland–Altman plot of estimated SBPs by Eq.(6) and ground-truth SBPs: (a) reference method with ECG and PPG, (b) proposed method with BCG and IPG.

Figure 7. Bland–Altman plot of estimated DBPs by Eq.(6) and ground-truth DBPs: (a) reference method with ECG and PPG, (b) proposed method with BCG and IPG.

Reviewer 2 Report

The paper describes blood pressure (BP) measurement using a ballistocardiogram (BCG) and impedance plethysmograph (IPG). The entire measurement system is integrated into a four-pads weight-fat scale. There is no need to install any sensor on the body, so the measurement of BP is easy, unobtrusive, and very practical. The method is based on a previous study by the same authors with some changes. The new method doesn’t use a PPG sensor on the toe, so it is entirely free of body-mounted sensors. On the other hand, the results are a little bit worse compared to the previous method, but the absence of body-mounted sensors is remarkable.

The presented paper generally has a clear aim and motivation, a good review of the state-of-the-art, well-described methods, and interesting results. English should be checked before the next submission. Although the article is well written, I have a few comments and suggestions below:

-       The schematic in Fig. 2 has very poor quality and is completely unreadable. Please enlarge the schematic to the full width of the page.

-       I would appreciate seeing the raw IPG signal and its processing. Could you provide images of IPG signal processing with comments?

-        In Fig. 3, some points (local maxima or local minima) are marked by black dots. Could you specify how these points were determined? Automatically or manually? If automatically, what method/methods were used?

Author Response

Reviewer 2 (round 1)

Dear Anonymous Reviewer,

The authors are grateful to your comments and suggestions for improving the quality and presentation of this paper. All comments are followed. The revised parts are highlighted in red. It is our sincere hope that this revision will enhance readability and strengthen of the manuscript to satisfy the requirements of this prestigious journal.

Comments and Suggestions for Authors

The paper describes blood pressure (BP) measurement using a ballistocardiogram (BCG) and impedance plethysmograph (IPG). The entire measurement system is integrated into a four-pads weight-fat scale. There is no need to install any sensor on the body, so the measurement of BP is easy, unobtrusive, and very practical. The method is based on a previous study by the same authors with some changes. The new method doesn’t use a PPG sensor on the toe, so it is entirely free of body-mounted sensors. On the other hand, the results are a little bit worse compared to the previous method, but the absence of body-mounted sensors is remarkable.

The presented paper generally has a clear aim and motivation, a good review of the state-of-the-art, well-described methods, and interesting results. English should be checked before the next submission. Although the article is well written, I have a few comments and suggestions below:

  1. The schematic in Fig. 2 has very poor quality and is completely unreadable. Please enlarge the schematic to the full width of the page.

ANS: Authors enlarges the size of Fig. 2 to make it easier for readers to read the text in the figure.

  1. I would appreciate seeing the raw IPG signal and its processing. Could you provide images of IPG signal processing with comments?

ANS: The raw and filtered IPG signals are shown in Fig. 4. But authors did a mistake for IPG and DIPG signals in Figure 3. The Fig. 3 has been modified. In order to let IPG signal more clear, Fig. 4 is added in the manuscript to show the raw and filtered IPG signals. The signal processing of IPG is described in “2.2 Digital Signal Processing” sector.

Line 150-176:

2.2 Digital Signal Processing

All measured signals are filtered to remove the wandering baseline and high frequency noise with the 4th-order Butterworth bandpass filter of the 0.5 Hz to 10 Hz bandwidth. To reduce differences of phase lag among different signals, an 8th-order all-pass filter was designed to equalize the group delay within the passband. Figure 3 shows the ECG (blue), PPG (red), differential PPG (DPPG, magenta), BCG (black), and IPG (green), and differential IPG (DIPG, purple). The PTT measured by BCG and IPG signals was defined as the interval between the J wave of BCG and the foot point of IPG for PTT1, and PTT2 is defined as the interval between the J wave of BCG and main peak of DIPG. The Pan and the Tompkins method was utilized to detect the R wave of ECG [48]. The first zero-crossing points of DIPG and DPPG after the R wave were defined as the foot-point times of IPG and PPG. The J wave of BCG is the first peak after the R wave. Then the first peaks of DIPG and DPPG are detected following their first zero-crossing points. In Figure 3, the R wave of ECG, J wave of BCG, and main peaks of DIPG and DPPG are marked by black dots, as are the foot points of IPG and PPG. Figure 4 shows the raw (top) and filtered (button) IPG signals.

Figure 3. The four signals and their differential signals, ECG (blue), PPG (red), differential PPG (DPPG, magenta), BCG (black), IPG (green), and differential IPG (DIPG, purple). The R wave of ECG, J wave of BCG, main peak of DPPG and DIPG, and foot point of PPG and IPG are marked by black dots. These signals are offset to visually separate them.

Figure 4. The raw (top) and filtered (bottom) IPG signals.

  1. In Fig. 3, some points (local maxima or local minima) are marked by black dots. Could you specify how these points were determined? Automatically or manually? If automatically, what method/methods were used?

ANS: Authors added some sentences in the first paragraph to describe how to detect the R wave of ECG, J wave of BCG, foot points of IPG and PPG, and peaks of DIPG and DPPG.

Line 150-165

2.2 Digital Signal Processing

All measured signals are filtered to remove the wandering baseline and high frequency noise with the 4th-order Butterworth bandpass filter of the 0.5 Hz to 10 Hz bandwidth. To reduce differences of phase lag among different signals, an 8th-order all-pass filter was designed to equalize the group delay within the passband. Figure 3 shows the ECG (blue), PPG (red), differential PPG (DPPG, magenta), BCG (black), and IPG (green), and differential IPG (DIPG, purple). The PTT measured by BCG and IPG signals was defined as the interval between the J wave of BCG and the foot point of IPG for PTT1, and PTT2 is defined as the interval between the J wave of BCG and main peak of DIPG. The Pan and the Tompkins method was utilized to detect the R wave of ECG [48]. The first zero-crossing points of DIPG and DPPG after the R wave were defined as the foot-point times of IPG and PPG. The J wave of BCG is the first peak after the R wave. Then the first peaks of DIPG and DPPG are detected following their first zero-crossing points. In Figure 3, the R wave of ECG, J wave of BCG, and main peaks of DIPG and DPPG are marked by black dots, as are the foot points of IPG and PPG. Figure 4 shows the raw (top) and filtered (button) IPG signals.

Reviewer 3 Report

The authors designed a system to estimate the noninvasive blood pressure from ballistocardiogram and impedance plethysmogram signals. Although the idea has a kind of novelty, the experiment design and method have to be improved to reach a certain level of scientific soundness.

1. As the ground truth data was attained from BP measurement device HM-7320, the electrical cuff BP device recommends measuring BP in a sitting position or lying down. However, the proposed experiment measured BP in a standing-up position. How could you make sure the ground truth data is reliable?

2. The topic presents " Touchless Measurement ".  Ballistocardiogram, impedance plethysmogram, ECG, and PPG are all needed to connect with the electrode attached on human body surface. How could you define "Touchless"?

3. There are three major standards to evaluate NBP measurement,  IEEE, AAMI, and BHS standards. Authors should refer to the other two standards.

4. Authors should state clear why the model to estimate NBP using ECG and PPG is needed in the proposed study.

5. Line 171, Pearson correlation coefficient, r2, describes the degree of linear regression. This sentence totally does not make sense. Formula (9) is not for the Pearson correlation coefficient.

6. If the study is about estimating the noninvasive blood pressure from ballistocardiogram and impedance plethysmogram signals, what is the relation between the study and weight-fat scale?

7. The results show the BP measurements from HM-7320 have a large-scale variation. The best practice to measure BP using HM-7320 or a similar device is to take the test three times and attain the average systolic and diastolic values.

Author Response

Reviewer 3 (round 1)

Dear Anonymous Reviewer,

The authors are grateful to your comments and suggestions for improving the quality and presentation of this paper. All comments are followed. The revised parts are highlighted in red. It is our sincere hope that this revision will enhance readability and strengthen of the manuscript to satisfy the requirements of this prestigious journal.

Comments and Suggestions for Authors

The authors designed a system to estimate the noninvasive blood pressure from ballistocardiogram and impedance plethysmogram signals. Although the idea has a kind of novelty, the experiment design and method have to be improved to reach a certain level of scientific soundness.

  1. As the ground truth data was attained from BP measurement device HM-7320, the electrical cuff BP device recommends measuring BP in a sitting position or lying down. However, the proposed experiment measured BP in a standing-up position. How could you make sure the ground truth data is reliable?

ANS: In this experiment, the cuff placement is the same level with the heart. According to the hydrostatic pressure theorem, if the placement of measured artery is the same level with heart, the hydrostatic pressure is zero. Moreover, subjects are stably standing on the weight. There is not the motion artifact for the BP measurement. But, the post-exercise BP decreases. That is, measured BP is dynamic. Accuracy of commercial BP monitor would decrease [53]. In the clinical practice, the static BP after resting 5 minutes represents the basic BP. The BP after the exercise does not represent the basic BP. In this study, the dynamic BP measured by the electrical BP device was the ground truth, which does not synchronize with the estimated BP. Thus, we have discussed this problem in “Discussions” sector.

Line 319-332:

Performance of BP measurement by the reference method did not reach that of previous studies [51]. In Table 5, the ERMSs of SBP and DBP measured by the reference method are only 6.0 ± 1.7 mmHg and 4.1 ± 1.6 mmHg; their r2s are 0.716 ± 0.162 and 0.439 ± 0.292. There are three main issues. First, the BP of subjects was raised by the exercise, which changes depend on interindividual differences. Thus, the maximum BP changes of SBP and DBP approached to 48 mmHg and 29 mmHg, and the minimum BP changes were 24 mmHg and 8 mmHg. Second, the post-exercise BP decreases. That is, measured BP is dynamic. Accuracy of commercial BP monitor would decrease [53]. Third, the reference BP was repeatedly measured once a minute, and it was difficult to synchronize the PTT. Therefore, for the proposed method, the ERMSs of SBP and DBP are 7.2 ± 2.2 mmHg and 4.4 ± 1.9 mmHg, the r2s are 0.618 ± 0.203 and 0.371 ± 0.275, as shown in Table 5. According to the standard developed by the Institute of Electrical and Electronics Engineers (IEEE) [54], accuracy of the proposed method reaches C grading. However, these results were very close to those of the reference method.

  1. The topic presents " Touchless Measurement ". Ballistocardiogram, impedance plethysmogram, ECG, and PPG are all needed to connect with the electrode attached on human body surface. How could you define "Touchless"?

ANS: Thanks for Reviewer 1 and 3 comment and suggestion. Authors changed the title of this manuscript as “Using Ballistocardiogram and Impedance Plethysmogram for Minimal Contact Measurement of Blood Pressure Based on a Body Weight-Fat Scale”.

  1. There are three major standards to evaluate NBP measurement, IEEE, AAMI, and BHS standards. Authors should refer to the other two standards.

ANS: Because the proposed method belongs to the wearable and cuffless BP measurement, authors used IEEE standard [49] as the reference. About the standards of AMMI, 60601-2-30, and BHS, BS EN 1060-4, these standards are used as the automatic cycling non-invasive blood pressure monitoring equipment. The BP device has to use the cuff to measure BP. Thus, in this study, authors did not use the standards of AMMI and BHS as the references.

[49].  IEEE Standard Association. IEEE standard for wearable, cuffless blood pressure measuring devices, IEEE Std. 2019.

  1. Authors should state clear why the model to estimate NBP using ECG and PPG is needed in the proposed study.

ANS: The cuffless BP measurement generally uses the ECG and PPG to get the PTT for BP estimation [22]. Thus, in this study, the ECG and PPG were used as the reference method to compare with the proposed method. Authors modified the sentences to state more clear in “Introduction” sector.

Line 65-77:

When the heart pumps the blood once, a blood pulse in the aorta transmits to the peripheral arterioles. This transmission time is called the pulse transit time (PTT) and has reciprocal relation with pulse wave velocity (PWV). Bramwell and Hill [19] proposed a model to explain the relation between blood pressure (BP) and PWV according to the Moens-Korteweg equation [20]. Because the blood pulse is caused by the left ventricular contraction, the R wave of the electrocardiogram (ECG) is usually used as the starting time of PTT. The foot point of a pulse wave of peripheral vessels is considered the ending time of PTT. Therefore, Sharwood-Smith proposed the use of PTT to estimate continuous systolic BP for monitoring its instantaneous drop under anesthesia [21]. In recent years, many studies have explored techniques for cuffless BP measurement [22], some of which have been implemented in wearable devices for eHealth [23,24]. Thus, in this study, the ECG and PPG are used to estimate BP as the reference method.

Line 103-117:

The body weight-fat scale and blood pressure monitor are the most popular apparatuses in eHealth [41–44]. Many studies show that there is a positive relationship between being overweight or obese and hypertension [45, 46]. Thus, daily measurements of body weight and BP are an important issue for self-care. To encourage the habit of self-monitoring, the apparatus should be easy and comfortable to use. Thus, the goal of this study is to explore minimal contact BP measurement based on a commercial body weight-fat scale when users stand on it. Users can easily measure their BPs by cuffless measurement. The PTT values were detected from BCG and IPG signals extracted from four strain gauges (SGs) and four footpads. The alternate current source for IPG measurement was supported by the commercial body weight-fat scale. To validate performance of the proposed method, the BP estimated by the reference method was compared with that of the proposed method. Four PPT models for estimating BP were used to explore the reliability and reproducibility of the proposed method.  The BP measurement system was integrated into a commercial body weight-fat scale. There is no need to install any sensor on the body, so measurement of BP is easy, unobtrusive, and very practical.

  1. Line 171, Pearson correlation coefficient, r2, describes the degree of linear regression. This sentence totally does not make sense. Formula (9) is not for the Pearson correlation coefficient.

ANS: Authors modified the description about the correlation coefficient, and Eq. (9).

Line 182-188

2.4 Statistic Analysis

The quantitative data is expressed as the mean ± standard deviation (SD). A two-tailed paired t-test is used to show the difference of two variables. A p-value of 0.05 or less is considered statistically significant. Cross-correlation coefficient, r2, is the quantity that gives the quality of a least squares fitting to the original data,  

.

(9)

Furthermore, the precision and agreement between the ground-truth BP and estimated BP by reference and proposed methods are compared using a Bland–Altman plot.

  1. If the study is about estimating the noninvasive blood pressure from ballistocardiogram and impedance plethysmogram signals, what is the relation between the study and weight-fat scale?

ANS: The BP measurement system is integrated into a commercial body weight-fat scale. There is no need to install any sensor on the body, so the measurement of BP is easy, unobtrusive, and very practical. Moreover, the alternate current source for the IPG measurement was supported by the commercial body weight-fat scale. Authors modified the texts in “Introduction” sector to state more clear.

Line 103-117:

The body weight-fat scale and blood pressure monitor are the most popular apparatuses in eHealth [41–44]. Many studies show that there is a positive relationship between being overweight or obese and hypertension [45, 46]. Thus, daily measurements of body weight and BP are an important issue for self-care. To encourage the habit of self-monitoring, the apparatus should be easy and comfortable to use. Thus, the goal of this study is to explore minimal contact BP measurement based on a commercial body weight-fat scale when users stand on it. Users can easily measure their BPs by cuffless measurement. The PTT values were detected from BCG and IPG signals extracted from four strain gauges (SGs) and four footpads. The alternate current source for IPG measurement was supported by the commercial body weight-fat scale. To validate performance of the proposed method, the BP estimated by the reference method was compared with that of the proposed method. Four PPT models for estimating BP were used to explore the reliability and reproducibility of the proposed method.  The BP measurement system was integrated into a commercial body weight-fat scale. There is no need to install any sensor on the body, so measurement of BP is easy, unobtrusive, and very practical.

  1. The results show the BP measurements from HM-7320 have a large-scale variation. The best practice to measure BP using HM-7320 or a similar device is to take the test three times and attain the average systolic and diastolic values.

ANS: In Table 1, Because the BP values are the dynamic BPs of subjects, the BPs measured by HM-7320 have a large-scale variation. The dynamic BP can not be averaged as the ground-truth BP. Thus, in this study, subjects were measured four times. The interval between two measurements was set longer than seven days intentionally. The experiment procedure is described in “2.5 Experiment Protocol” sector.

Line 205-214:

1. Subjects stand on the body weight-fat scale to measure ECG, PPG, IPG, and BCG signals for five minutes, and they measure BP once as a baseline.

2. Subjects run on a treadmill at a speed of about 6 km/h for at least three minutes, and 8 km/h for the next four minutes. If the SBP is not raised to 20 mmHg higher than the resting SBP, subjects are requested to run longer.

3. Subjects stand on the commercial body weight-fat scale again, measuring ECG, PPG, IPG, and BCG signals for six minutes. The BP is measured once a minute when standing on the body weight-fat scale.

4. Each measurement session requires about 18 minutes. Subjects are measured four times. The interval between any two measurement sessions is at least a week.

Round 2

Reviewer 1 Report

Abstract line 26, the abbreviation for root-mean-square errors is typically RMSEs.

Line 159, “The Pan and the Tompkins method…” should simply be, “The Pan-Tompkins method…”

Lines 75 and 76 state, “Thus, in this study, the ECG and PPG are used…” This is most likely a typo since IPG is used, not PPG.

Even with the updates to the Introduction, it is still not clear what the novelty of this manuscript is compared to previous studies.

The text in Figure 2 is still too small to be easily readable. The authors could consider breaking Figure 2 up into multiple figures.

Author Response

Reviewer 1 (round 2)

Dear Anonymous Reviewer,

The authors are grateful to your comments and suggestions for improving the quality and presentation of this paper. All comments are followed. The revised parts are highlighted in red. It is our sincere hope that this revision will enhance readability and strengthen of the manuscript to satisfy the requirements of this prestigious journal.

Comments and Suggestions for Authors

1. Abstract line 26, the abbreviation for root-mean-square errors is typically RMSEs.

ANS: Authors corrected this mistake. We deleted this abbreviation.

Line 26:

By the proposed model, the root-mean-square errors and correlation coefficients (r2s) of estimated systolic blood pressure and diastolic blood pressure are 7.3 ± 2.1 mmHg and 4.5 ± 1.8 mmHg, and 0.570 ± 0.205 and 0.284 ± 0.166, respectively.

2. Line 159, “The Pan and the Tompkins method…” should simply be, “The Pan-Tompkins method…”

ANS: Authors modified this sentence.

Line 159:

The Pan-Tompkins method was utilized to detect the R wave of ECG [48].

3. Lines 75 and 76 state, “Thus, in this study, the ECG and PPG are used…” This is most likely a typo since IPG is used, not PPG.

ANS: According to the comment of Reviewer 3, authors explain why this study used ECG and PPG signals as the “reference” method for the BP estimation. Thus, this sentence is right.

4. Even with the updates to the Introduction, it is still not clear what the novelty of this manuscript is compared to previous studies.

ANS: There are three contributions in this study. First, The BP measurement system was integrated into a commercial body weight-fat scale. The driving  current source for IPG measurement was supported by the body weight-fat scale. Second, the circuit for the IPG measurements were supported in the method sector. Third, the accurate differences of cuffless BP measured by the proposed and reference methods were explored. The delay time between PPG and IPG on the finger and toe was analyzed. Authors modified the last paragraph in Introduction sector to let readers understand the contribution of this study.     

Line 107-119:

Thus, the goal of this study is to explore minimal contact BP measurement based on a commercial body weight-fat scale when users stand on it. The PTT values were detected from BCG and IPG signals extracted from four strain gauges (SGs) and four footpads. Four PPT models for estimating BP were used to explore the reliability and reproducibility of the proposed method. There are three contributions in this study. First, The BP measurement system was integrated into a commercial body weight-fat scale. The driving  current source for IPG measurement was supported by the body weight-fat scale. Second, the circuit for the IPG measurements were supported in the method sector. Third, the accurate differences of cuffless BP measured by the proposed and reference methods were explored. The delay time between PPG and IPG on the finger and toe was analyzed. Thus, the proposed method is no need to install any sensor on the body, so measurement of BP is easy, unobtrusive, and very practical.

5. The text in Figure 2 is still too small to be easily readable. The authors could consider breaking Figure 2 up into multiple figures.

ANS: Authors modified the size of texts in Fig. 2 to let readers easily read the information of IPG schematic circuit.

Reviewer 3 Report

The authors addressed the questions raised before.

Author Response

Reviewer 3 (round 2)

Dear Anonymous Reviewer,

The authors are grateful to your comments and suggestions for improving the quality and presentation of this paper. All comments are followed. The revised parts are highlighted in red. It is our sincere hope that this revision will enhance readability and strengthen of the manuscript to satisfy the requirements of this prestigious journal.

Comments and Suggestions for Authors

The authors addressed the questions raised before.

ANS: Thanks for reviewer’s comments.